# Measuring the unknown: An estimator and simulation study for assessing case reporting during epidemics

Christopher I. Jarvis[1,2]*, Amy Gimma[1,2], Flavio Finger[1,2,3], Tim P. Morris[4], Jennifer A. Thompson[1], Olivier le Polain de Waroux[2], W. John Edmunds[1,2], Sebastian Funk[1,2], Thibaut Jombart[1,2,5,6]

1 Department of Infectious Disease Epidemiology, London School of Hygiene & Tropical Medicine, London, United Kingdom, 2 Centre for Mathematical Modelling of Infectious Diseases, London School of Hygiene & Tropical Medicine, London, United Kingdom, 3 Epicentre, Paris, France, 4 MRC Clinical Trials Unit at UCL, London, United Kingdom, 5 MRC Centre for Global Infectious Disease Analysis, Department of Infectious Disease Epidemiology, School of Public Health, Imperial College London, London, United Kingdom, 6 UK Public Health Rapid Support Team, London, United Kingdom

* christopher.jarvis@lshtm.ac.uk

**Data Availability Statement:** This paper uses only simulated data. All scripts for generating simulated datasets and performing subsequent analyses are

## Abstract

The fraction of cases reported, known as 'reporting', is a key performance indicator in an outbreak response, and an essential factor to consider when modelling epidemics and assessing their impact on populations. Unfortunately, its estimation is inherently difficult, as it relates to the part of an epidemic which is, by definition, not observed. We introduce a simple statistical method for estimating reporting, initially developed for the response to Ebola in Eastern Democratic Republic of the Congo (DRC), 2018–2020. This approach uses transmission chain data typically gathered through case investigation and contact tracing, and uses the proportion of investigated cases with a known, reported infector as a proxy for reporting. Using simulated epidemics, we study how this method performs for different outbreak sizes and reporting levels. Results suggest that our method has low bias, reasonable precision, and despite sub-optimal coverage, usually provides estimates within close range (5–10%) of the true value. Being fast and simple, this method could be useful for estimating reporting in real-time in settings where person-to-person transmission is the main driver of the epidemic, and where case investigation is routinely performed as part of surveillance and contact tracing activities.

## Author summary

When responding to epidemics of infectious diseases, it is essential to estimate how many cases are not being reported. Unfortunately reporting, the proportion of cases actually observed, is difficult to estimate during an outbreak, as it typically requires large surveys to be conducted on the affected populations. Here, we introduce a method for estimating reporting from case investigation data, using the proportion of cases with a known, reported infector. We used simulations to test the performance of our approach by

available on github repositories as indicated in the manuscript and available here https://github.com/jarvisc1/2020-reporting.

**Funding:** TJ, CIJ and WJE receive funding from the Global Challenges Research Fund (GCRF) project 'RECAP' managed through RCUK and ESRC (ES/P010873/1). TJ and OPW receive funding from the UK Public Health Rapid Support Team funded by the United Kingdom Department of Health and Social Care. TJ receives funding from the National Institute for Health Research - Health Protection Research Unit for Modelling Methodology. TPM was funded by the UK MRC (grants MC_UU_12023/21 and MC_UU_12023/29). JAT was jointly funded by the UK Medical Research Council (MRC) and the UK Department for International Development (DFID) under the MRC/DFID Concordat agreement and is also part of the EDCTP2 programme supported by the European Union (MR/R010161/1) The UK Public Health Rapid Support Team is funded by the United Kingdom Department of Health and Social Care. The views expressed in this publication are those of the authors and not necessarily those of the National Health System, the National Institute for Health Research or the Department of Health and Social Care. The authors alone are responsible for the views expressed in this article and they do not necessarily represent the views, decisions or policies of the institutions with which they are affiliated. The funders had no role in study design, data collection and analysis, decision to publish, or preparation of the manuscript.

**Competing interests:** The authors have declared that no competing interests exist.

mimicking features of a recent Ebola epidemic in the Democratic Republic of the Congo. We found that despite some uncertainty in smaller outbreaks, our approach can be used to obtain informative ballpark estimates of reporting under most settings. This method is simple and computationally inexpensive, and can be used to inform the response to any epidemic in which transmission events can be uncovered by case investigation.

This is a *PLOS Computational Biology* Methods paper.

## Introduction

The response to infectious disease outbreaks increasingly relies on the analysis of various data sources to inform operation in real time [1,2]. Outbreak analytics can be used to characterise key factors driving epidemics, such as transmissibility, severity, or important delays like the incubation period or the serial interval [2]. Amongst these factors, the amount of infections remaining undetected in the affected populations is a crucial indicator for assessing the state of an epidemic, and yet this quantity is often hard to estimate in real time [3–6]. Indeed, estimation of the overall proportion of individuals infected (attack rates) typically requires time-consuming serological surveys [7–9] which may not be achievable in resource-limited, large-scale emergencies such as the 2014–2016 Ebola virus disease (EVD) outbreak in West Africa [10], or the more recent EVD outbreak in Eastern provinces of the Democratic Republic of the Congo (DRC) [11,12].

As an alternative, one may attempt to quantify *reporting*, i.e. the proportion of all infections which result in notified cases. Unfortunately, this quantity is also hard to estimate, and usually requires the analysis of epidemiological and genomic data through complex methods for reconstructing transmission trees [13–15] or transmission clusters [16]. Such requirements can mean that by the time estimates are available, decisions have already been made, or the outbreak situation has changed [17–19]. Therefore, simpler approaches are needed for estimating reporting and help inform outbreak response operations.

Methods for estimating reporting during an outbreak should ideally exploit data which is routinely collected as part of the outbreak response. In diseases where dynamics are mostly governed by person-to-person transmission, case investigation and contact tracing can be powerful tools for understanding past transmission events as well as detecting new cases as early as possible [11,20–23]. For vaccine-preventable diseases, contact tracing can also be used for designing ring vaccination strategies, as seen in recent EVD outbreaks in the DRC [11,20]. These data also contain information about reporting. Intuitively, the frequency of cases whose infector is a known and reported case is indicative of the level of reporting: the more frequently case investigation identifies a known infector, the higher the corresponding case reporting should be. Conversely, cases with no known epidemiological link after investigation are indicative of unobserved infections, and therefore under-reporting.

In this article, we introduce a method to estimate case reporting from contact tracing data. This approach, designed during the Ebola outbreak in Eastern DRC [11,12], was originally aimed at assessing case reporting in a context where insecurity made surveillance difficult, and under-reporting likely [12]. The approach utilized transmission chain data and calculated the proportion of cases with a known epidemiological link as a proxy for reporting. We provide a derivation of the estimator and explain the rationale of this approach and assess its performance using simulated outbreaks of different sizes with varying levels of reporting. Based on the simulation results, we make some suggestions regarding the use of this method to inform strategic decision making during an outbreak response.

## Methods

We present the analytical derivation of our method of estimating reporting, defined as the proportion of cases actually notified during an outbreak. We then describe the simulation study, using the ADEMP (Aim, Data generating mechanism, Estimand, Methods, Performance measures) framework as described by Morris et al 2019 [24,25], used to evaluate the performance of the methods under various conditions.

### Estimating reporting from epidemiological links

Our method exploits transmission chains derived from case investigation and contact tracing data. The data considered are secondary cases for which epidemiological investigation was successfully carried out, and for which a single likely infector could be clearly identified. We thus distinguish i) cases for which the identified infector is listed amongst reported cases (*cases with a known infector*) and ii) cases for which the identified infector is not listed amongst the reported cases (*cases without a known infector*). Importantly, cases without any known exposure, or cases for which multiple epidemiological links make it hard to disentangle a single likely infector, are excluded from the analysis.

The rationale for the approach is to consider the proportion of cases with a known infector as a proxy for the proportion of infections (including asymptomatic but infectious individuals) reported. The proportion of cases with a known infector is by definition the proportion of infectors who were reported (Fig 1), so that the reporting probability $\pi$ can be estimated as $\hat{\pi} = \frac{n_k}{n_k + n_u}$ where $n_k$ is the number of secondary cases (infectees) with a known infector and $n_u$ is the number of secondary cases without a known infector.

### Derivation of estimator for reporting

We define

$m_r$—number of reported infectors

$m_u$—number of unreported infectors

$n_k$—number of secondary cases (infectees) with known infector

$n_u$—number of secondary cases without known infector

$R$—reproduction number, i.e. average number of secondary cases by case; we assume reported and unreported infectors have the same distribution of $R$

$\pi$—reporting probability following some unspecified probability distribution with unknown probability parameter such that $E(\pi) = \frac{m_r}{m_r + m_u}$ where secondary cases are assumed to follow the same reporting distribution as primary infections.

The expected number of reported infectees with a known infector is

$$E(n_k) = m_r \, R\pi.$$

Similarly, the expected number of reported infectees without a known infector is

$$E(n_u) = m_u \, R\pi$$

From this we have that

$$m_r = \frac{E(n_k)}{R\pi} \;\; and \;\; m_u = \frac{E(n_u)}{R\pi}$$

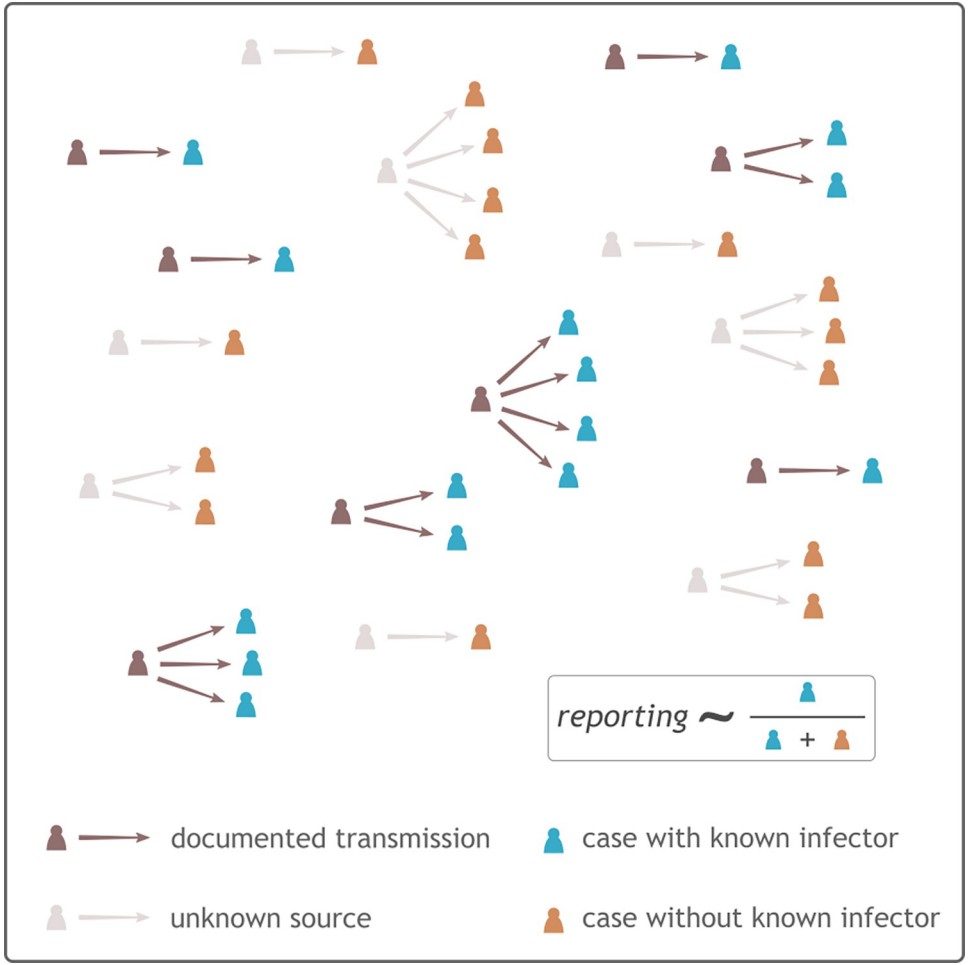

**Fig 1. Rationale of the method for estimating reporting.** This diagram illustrates transmission events inferred by case investigation of reported secondary cases, with arrows pointing from infectors to infectees. Darker shades are used to indicate documented transmission events, while lighter shades show unknown infectors. Numbers of secondary cases with (blue) or without (orange) known infectors are used to estimate the reporting probability. This example uses an approximate reporting of 50%.

By definition

$$\pi = \frac{m_r}{m_r + m_u}$$

Therefore

$$E(\pi) = \frac{\frac{E(n_k)}{R\pi}}{\frac{E(n_k)+E(n_u)}{R\pi}} = \frac{E(n_k)}{E(n_k) + E(n_u)}$$

and replacing the expectations with their estimates from the data, we get the estimator

$$\hat{\pi} = \frac{n_k}{n_k + n_u}.$$

## Uncertainty for reporting

The uncertainty associated with this estimation can be estimated using various methods for computing confidence intervals of proportions. Using the standard approach for standard errors for a proportion we have that

$$SE_\pi = \frac{\hat{\pi} \times (1 - \hat{\pi})}{n_k + n_u}.$$

Here, we used exact binomial confidence intervals which can be calculated:

$$(1 + \frac{n - n_k + 1}{n_k F[\frac{\alpha}{2}; \; 2n_k, 2(n - n_k + 1)]})^{-1} < \pi < (1 + \frac{n - n_k}{(n_k + 1)F[1 - \frac{\alpha}{2}; \; 2(n_k + 1), 2(n - n_k)]})^{-1}$$

Where $n = n_k + n_u$, total number of secondary cases. $F(c; d_1, d_2)$ is the c quantile from an F-distribution with $d_1$, $d_2$ degrees of freedom and $1-\alpha$ is the confidence level.

## Simulation study

**Aim.**   We aim to test the performance of the method for different outbreak sizes and actual reporting, in terms of bias, coverage, and precision (in an operational context) using simulated outbreaks.

**Data generating mechanism.**   We considered twelve data-generating mechanisms (three reporting rates by four reported outbreaks sizes) and performed 4000 repetitions per mechanism.

Each repetition corresponded to a hypothetical outbreak with a known transmission tree. To simulate the reporting process, cases were removed randomly from the transmission chains using a Binomial process with a probability (1—*reporting*). We will thus distinguish the *total outbreak size*, which represents all cases in the outbreak, and the *reported outbreak size*, *which* represents the number of cases reported. For simplicity, we assumed that all cases reported were investigated, so that it is known if they had a documented epidemiological link, or not, amongst reported cases.

For each outbreak (repetition) we removed observations so that reporting was 25%, 50%, or 75%. Therefore a single simulated outbreak will give three different observed outbreaks. We categorised the simulations into reported outbreak sizes of 1–99, 1–499, 500–999, 1000+.

## Outbreak simulation

We used the R package *simulacr* [26] to simulate outbreaks, the reporting process, and the subsequently observed transmission chains. *simulacr* implements and extends individual-based simulations of epidemics previously used to evaluate transmission tree reconstruction methods [13,14,27]. In its basic form, *simulacr* implements a Poisson branching process in which the reproduction numbers (*R*) is combined with the infectious period to determine individual rates of infection. Here, to account for potential heterogeneity in transmission, we have drawn individual values of *R* from a Gamma distribution fitted to empirical data from the North Kivu EVD epidemic (rate: 1.2; shape: 2; corresponding mean: 1.7). The resulting branching process being a combination of Poisson processes with Gamma-distributed rates is therefore a Negative Binomial branching process. The infectiousness of a given individual *i* at time *t* is, noted $\lambda_{i,t}$, is calculated as:

$$\lambda_{i,t} = R_i \text{w}(t - s_i)$$

where $R_i$ is the reproduction number for individual *i*, $s_i$ is their date of symptom onset, and w

is the probability mass function of the duration of infectiousness (time interval between onset of symptom and new secondary infections). New cases generated at time $t+1$ are drawn from a Poisson distribution with a rate $\Lambda_t$ summing the infectiousness of all cases:

$$\Lambda_t = (n_s/n) \sum_i \lambda_{i,t}$$

where $n_s$ is the number of susceptible individuals and $n$ the total population size, so that the branching process includes a density-dependence in which rates of infection decrease with the proportion of susceptibles.

Transmission trees are built by assigning infectors to newly infected individuals according to a multinomial distribution in which potential infectors have a probability $\lambda_{i,t} / \sum_i \lambda_{i,t}$ of being drawn. The dates of symptom onset and case notification are generated for each new case using user-provided distributions for the incubation time and reporting delays. Simulations run until any of the set duration of the simulation is reached (here, 365 days).

Here, we used parameters values and distributions in line with estimates from the Eastern DRC Ebola outbreak [12,28], the details of which are provided in Table 1. All code used for running these simulations is available from https://github.com/jarvisc1/2020-reporting.

**Estimand: Reporting.**   We considered a single estimand $\pi$ the level of reporting.

## Method

For each repetition we calculated the proportion of the number of cases with a known infector over the total number of reported cases, that is the estimator $\hat{\pi} = \frac{n_k}{n_k+n_u}$. We further calculated the standard error and 95% exact binomial confidence intervals.

### Performance measures

The performance of the method was measured using bias, coverage, and precision. For bias and coverage, the Monte-Carlo standard errors were calculated to quantify uncertainty about the estimates of the performance [29]. The equations used are detailed in Table 2 and were taken from Morris et al [24]. In addition, results were classified according to different ranges of *absolute error*, for a more operational interpretation of the results.

**Table 1. Parameters used for simulating outbreaks.** This table details input parameters used for simulating outbreaks using the R package *simulacr*. Fixed values were used for all simulations, and reflect the natural history of the 2018–2020 Eastern DRC Ebola outbreak. Variable values changed across simulations.

| **Fixed values** | |
| --- | --- |
| Maximum duration of the outbreak | 365 days |
| Incubation time distribution | Discretised gamma distribution<br>mean of 9.7 days, sd = 5.5 days. |
| Infectious period distribution | Discretised gamma distribution<br>mean = 5 days, sd = 4.7 days. |
| Reproduction number distribution | Gamma distribution:<br>rate of 1.2 shape of 2. |
| **Variable values** | |
| Population size* | 200, 500, 1000, 2000, 5000, 7500, 10000, 15000, 20000 |
| Outbreak size* | 10–99, 100–499, 500–999, 1000+ |
| Proportion of cases not reported | 0.25, 0.50, 0.75 |

*Population size is controlled in each simulation, the outbreak sizes are determined after the outbreaks have been simulated and the proportion of cases not reported have been removed.

**Table 2. Metrics used to measure performance in the simulation study.**

| Performance measure | Definition |
|---|---|
| Bias | $\delta = E[\hat{\theta}] - \theta$ where $\theta$ is the true value and $\hat{\theta}$ is the estimate of value |
| Coverage | If we define a confidence interval $(\hat{\theta}_{low}, \hat{\theta}_{upp})$ as the $P(\hat{\theta}_{low} \leq \theta \leq \hat{\theta}_{upp}) = \psi$ where $\psi \in [0, 1]$ then a 95% CI is when $P(\hat{\theta}_{low} \leq \theta \leq \hat{\theta}_{upp}) = 0.95$. It follows that coverage is the $P(\hat{\theta}_{low} \leq \theta \leq \hat{\theta}_{upp})$. |
| Precision | |
| Model based standard error | $\sqrt{E[\hat{Var}(\hat{\theta})]}$ |
| Empirical based standard error | $\sqrt{Var(\hat{\theta})}$ |
| Absolute error | $\|\hat{\theta}_i - \theta\|$ |

*Bias* is the difference between the expected value and the true value. It was measured by taking the difference between the average estimate of reporting versus the true reporting. Unbiasedness is a desirable statistical quality but a small amount of bias may be tolerated in exchange for other desirable qualities of an estimator. The estimates of reporting were presented visually by displaying the estimates of all 4000 simulations for each scenario.

*Coverage* is the percentage of CIs containing the true value. In the case of a 95% CI this should contain the true value 95% of the time. We counted the number of repetitions where the true value was contained in the 95% CI and divided by the total number of repetitions. The coverage was visualised through the use of Zip plots. This new visualisation created by Morris et al [24], helps to assess the coverage of a method by viewing the CIs directly. Assessing an expected 95% coverage with a Monte-Carlo standard error of 0.35 requires 3877 simulations [24] which is well within our 4000 simulations.

*Precision* represents how close the estimates are to each other. The model-based and empirical standard error were also calculated to provide an indication of precision. The model based standard error is the root of the mean estimated variance, and the empirical standard error represents the spread of the estimates. This gives an indication of how much the point estimates vary across simulations based on the level of reporting and sample size. Although the method may give unbiased estimates with good coverage under repeated sampling, an imprecise method could lead to large differences from the true value when applied to a single dataset (that is, confidence intervals may cover the true value honestly but are wide).

We further explored the impact of bias and *precision* of the estimator by considering the deviations of the estimates from the true value termed *absolute error*. The *absolute error* is defined as the absolute difference between the estimated reporting and its true value, expressed as percentages. For instance, estimates of 43% and 62% for a true reporting of 50% would correspond to absolute errors of 7% and 12%, respectively. During a disease outbreak, decisions are frequently made in the face of large uncertainties, and small absolute differences in the estimated level of reporting are unlikely to result in strategic changes. Therefore, as a perhaps more operationally relevant metric, we categorised results according to how far from the true value estimates were, using an arbitrary scale: very close ($\leq$5% absolute error), close ($\leq$10%), approximate ($\leq$15%) or inaccurate ($\leq$20%).

## Sensitivity to *R* values

In order to explore the sensitivity of the method to the distribution of *R*, we repeated the simulations and analyses using different distributions of the reproduction number, using Gamma

(rate = 0.95, shape = 2) and Gamma(rate = 1.475, shape = 2), resulting in average *R* values of 2.1 and 1.3, respectively, broadly in line with values reported in the literature for other EVD outbreaks [30].

## Results

### Bias

There was very little bias across all the simulated scenarios (Table 3 and Fig 2). For outbreaks with over 100 cases all estimates of bias were 0 with decreasing Monte Carlo error from 0.04 to 0.01 as the size of the reported outbreak increased. For outbreaks reported as less than 100 cases the bias was -0.1 for reporting of 0.50 and 0.75 and 0 for 0.25 with Monte Carlo error of 0.07. Table 3 presents the bias for each scenario and it can be seen that all of these estimates were within one standard error from zero, suggesting reasonable confidence that this is an overall unbiased estimator.

### Coverage

The coverage varied across the simulated scenarios with all but reported outbreak size 10–99 with reporting at 0.25 displaying under-coverage (Fig 3). The coverage was poor with all coverage estimates more than one standard error away from 95%, and most several standard errors away (Table 3). There was some suggestion of the counterintuitive pattern that coverage decreased as the reporting increased and that coverage decreased as the outbreak size increased.

### Precision

The model based standard error was below 0.07 for all estimates and below 0.04 for reported outbreaks of over 100 cases. Similar patterns are seen for the empirical standard error. Imprecise estimates were most marked when reported outbreaks were less than 100 cases and had 0.75 reporting. The precision increased (model based and empirical standard error decreased) as the reported outbreak size increased (Fig 2 and Table 3). Overall the precision appears reasonable when outbreaks are larger than 100.

### Absolute error

Results showed that the estimates were rarely more than 15% away from the true reporting value in all simulation settings (Fig 4 and Table 4). The absolute error was negligible in all

**Table 3. Performance measures from 4000 simulation by reported outbreak size and true reporting level.** Estimate (Monte-carlo standard error).

| Performance measures (MCSE) | Proportion reported | Reported outbreak size | | | |
|---|---|---|---|---|---|
| | | 10–99 | 100–499 | 500–999 | 1000 or more |
| **Bias** | 0.25 | 0 (0.07) | 0 (0.03) | 0 (0.02) | 0 (0.01) |
| | 0.5 | -0.01 (0.07) | 0 (0.04) | 0 (0.02) | 0 (0.01) |
| | 0.75 | -0.01 (0.07) | 0 (0.04) | 0 (0.02) | 0 (0.01) |
| **Coverage** | 0.25 | 95.7% (0.3) | 94.1% (0.4) | 94.4% (0.4) | 93% (0.4) |
| | 0.5 | 92.6% (0.4) | 92.4% (0.4) | 91.3% (0.4) | 91.2% (0.4) |
| | 0.75 | 92.3% (0.4) | 91.5% (0.4) | 89.2% (0.5) | 88.6% (0.5) |
| **Model standard error** | 0.25 | 0.065 (0) | 0.024 (0) | 0.015 (0) | 0.01 (0) |
| | 0.5 | 0.061 (0) | 0.038 (0) | 0.019 (0) | 0.011 (0) |
| | 0.75 | 0.059 (0.001) | 0.036 (0) | 0.014 (0) | 0.011 (0) |
| **Empirical standard error** | 0.25 | 0.071 (0.001) | 0.025 (0) | 0.016 (0) | 0.01 (0) |
| | 0.5 | 0.07 (0.001) | 0.044 (0) | 0.022 (0) | 0.012 (0) |
| | 0.75 | 0.068 (0.001) | 0.043 (0) | 0.017 (0) | 0.013 (0) |

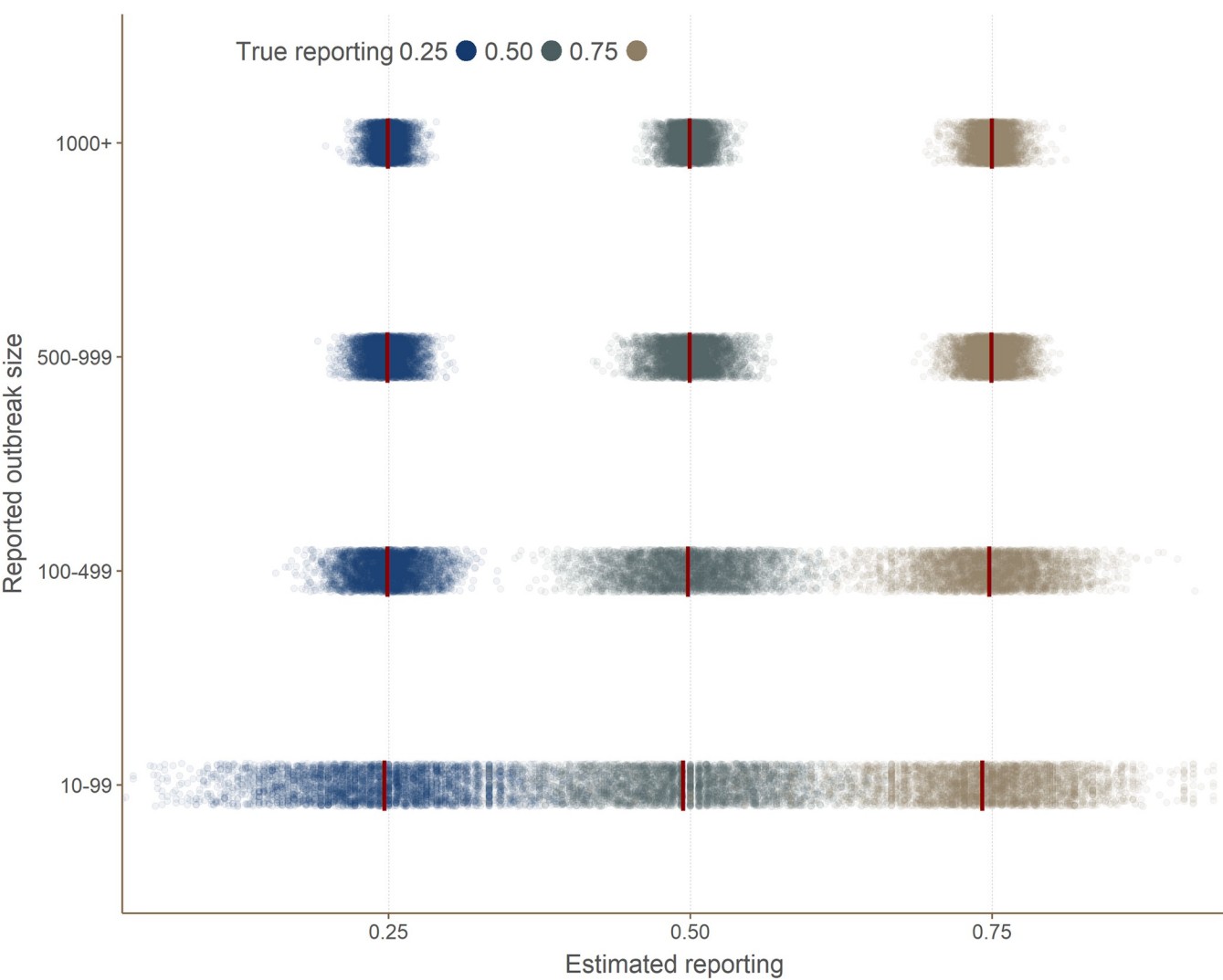

**Fig 2. Comparison of estimated versus actual reporting.** This graph shows the results of reporting estimated by the method for 4000 simulated outbreaks, broken down by outbreak size category (y-axis). Each dot corresponds to an independent simulation. The vertical red bars indicate the average within each category. True reporting used in the simulations is indicated by colors.

larger reported outbreaks (500 cases and above), with nearly all estimates very close (within 5%) to the true reporting value. Performance decreased in smaller outbreaks, but most estimates remained close (within 10%) to the true value. Results were worse in smaller outbreaks (10–99 reported cases), but even there about half of the estimates were very close (within 5%) to the true value, and more than 80% of estimates were within 10% of the target.

## Sensitivity to *R* values

Repeating the analyses with different *R* distributions in line with reproduction numbers reported in other outbreaks [30] showed negligible impact of *R* on bias, coverage, precision, and absolute error (Tables A and B in S1 Text). Coverage was the most sensitive to the change in *R*, which slightly decreased with higher mean *R* values. Overall though, variations were negligible compared to variations of coverage with epidemic size.

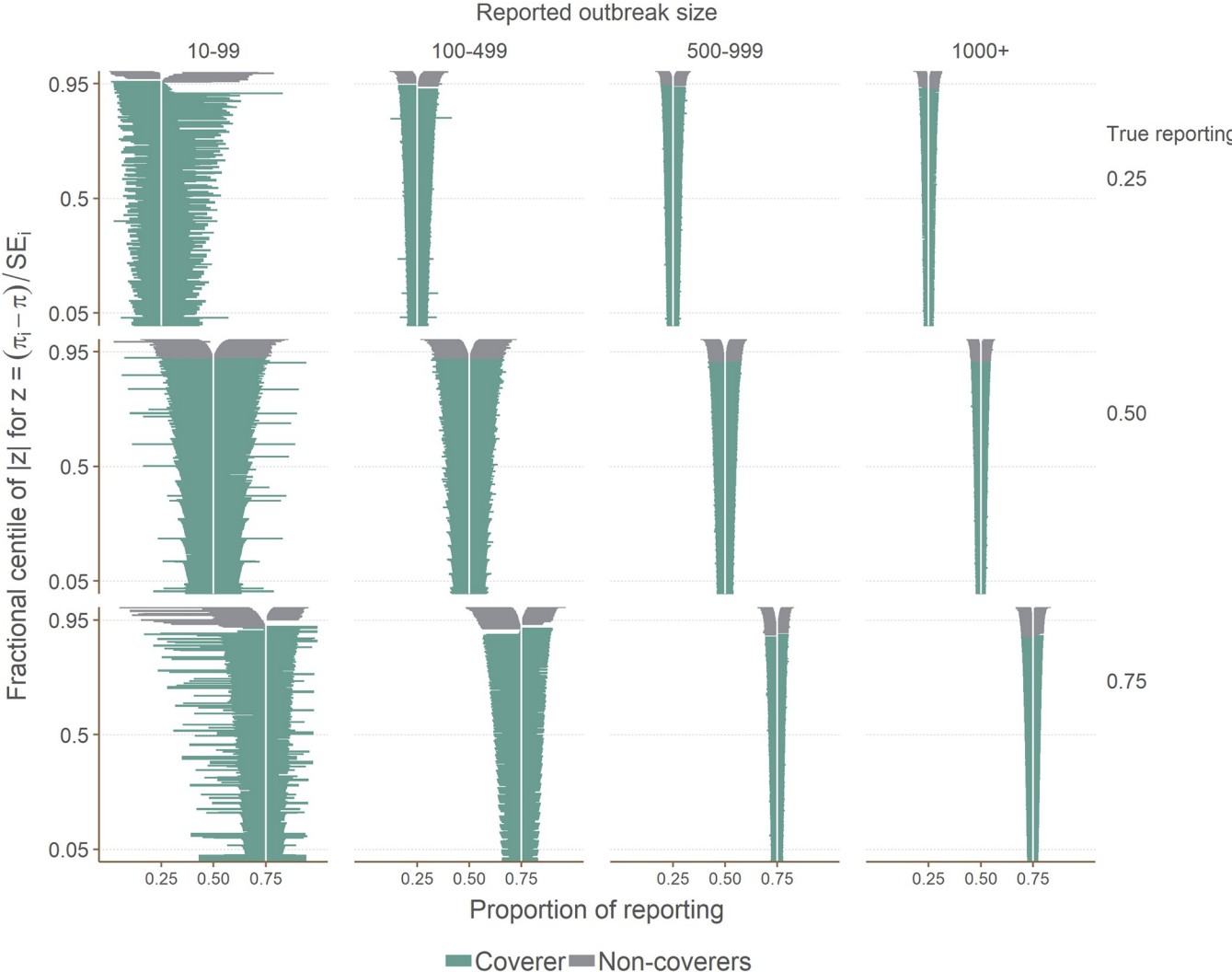

**Fig 3. Zip plot of showing coverage results.** This graph shows the 95% confidence intervals estimated by the method, broken down by reported outbreak size category and true reporting value. The vertical axis represent the fractional centile of $|Z|$ where $Z = \frac{(\pi_i - \pi)}{SE_i}$ and $\pi$ is reporting. The confidence intervals are ranked by their level of coverage and thus the vertical axis can be used to determine the proportion of confidence intervals that contain the true value where 0.95 would represent a coverage of 95%.

## Discussion

We have presented a new estimator for the levels of reporting in an outbreak based on the proportion of cases with known infectors, which can be derived from case investigation data. Using simulated outbreaks to assess the performance of the method, we found that this approach generally had little bias, reasonable precision, but poor coverage. Across all simulations, estimated reporting was most often within 10% of the true value, suggesting the method will retain operational relevance under different settings. The results were not sensitive to the range of reproduction numbers simulated in the scenarios, suggesting that the method can be applied in settings of somewhat higher and lower transmission.

Simulation results indicate a first limitation of the method lies in the analysis of smaller outbreaks. Overall, the approach performed better in larger outbreaks, with all metrics pointing to improved results in outbreaks of more than 100 case investigations. This observation suggests

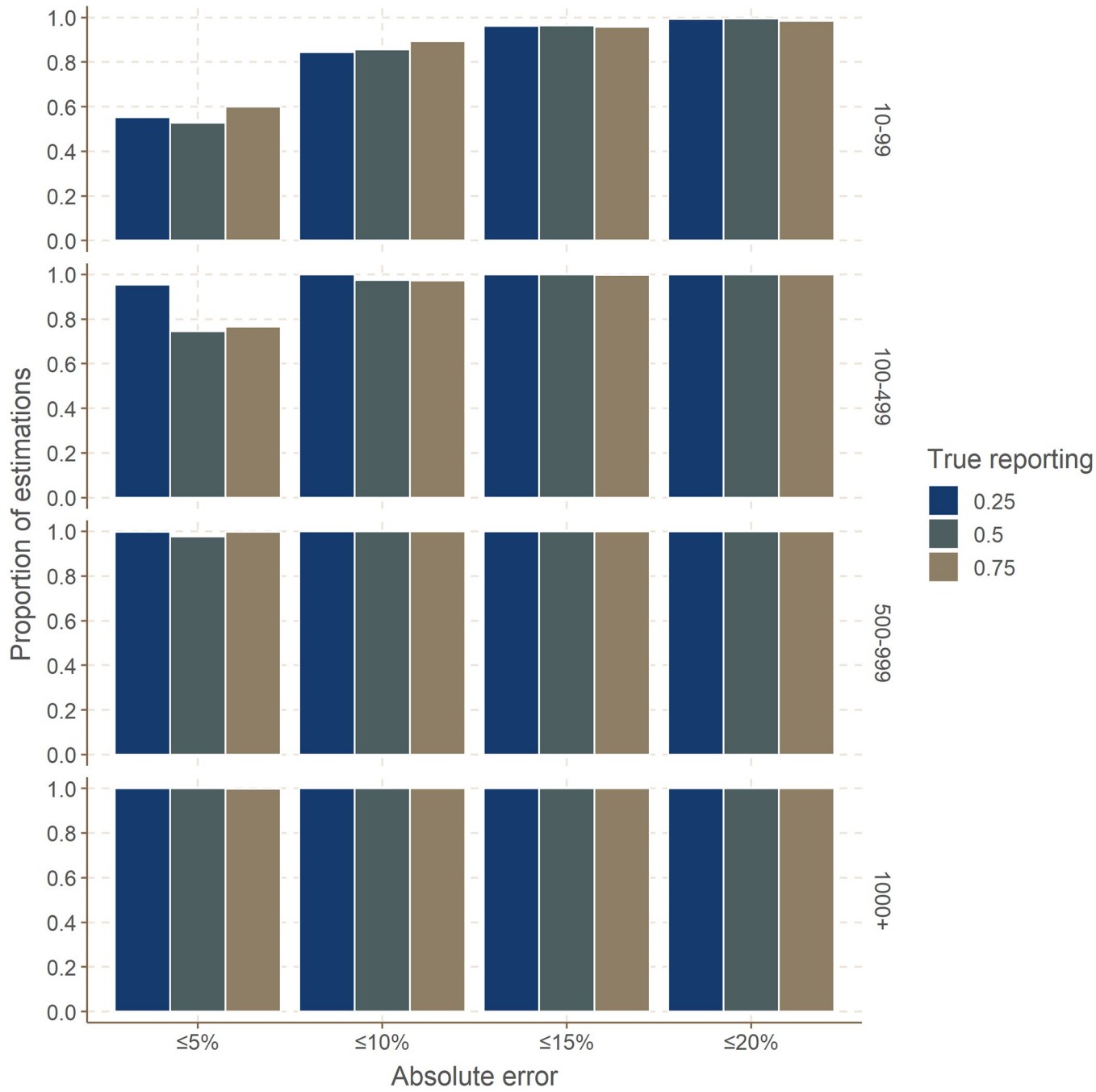

**Fig 4. Absolute error in reporting estimation.** This graph shows, for different simulation settings, the proportion of results within a given margin of absolute error, expressed as the absolute difference between the true and the estimated reporting (in %). Rows correspond to different outbreak size categories (outbreak size as reported). True reporting is indicated in color.

that our method may struggle to identify heterogeneities in reporting across time, space, or sections of the population (*e.g.* age groups) if the corresponding strata have small numbers of reported cases. It also means that estimates of reporting made in the early stages of an outbreak, when few cases have been reported and investigated, will be prone to larger statistical uncertainty.

**Table 4. Comparison of absolute error from 4000 simulations between true reporting levels and estimate of reporting by reported outbreak size and true reporting level.**

| Proportion reported | Reported outbreak size | Absolute error from true value | | | |
|---|---|---|---|---|---|
| | | ≤ 5% | ≤ 10% | ≤ 15% | ≤ 20% |
| 0.25 | 10–99 | 2213 (55.3%) | 3376 (84.4%) | 3849 (96.2%) | 3973 (99.3%) |
| | 100–499 | 3817 (95.4%) | 4000 | 4000 | 4000 |
| | 500–999 | 3995 (99.9%) | 4000 | 4000 | 4000 |
| | 1000+ | 3999 (100%) | 4000 | 4000 | 4000 |
| 0.5 | 10–99 | 2110 (52.8%) | 3430 (85.8%) | 3860 (96.5%) | 3978 (99.4%) |
| | 100–499 | 2981 (74.5%) | 3899 (97.5%) | 3998 (100%) | 4000 |
| | 500–999 | 3905 (97.6%) | 4000 | 4000 | 4000 |
| | 1000+ | 4000 | 4000 | 4000 | 4000 |
| 0.75 | 10–99 | 2400 (60%) | 3575 (89.4%) | 3835 (95.9%) | 3942 (98.6%) |
| | 100–499 | 3067 (76.7%) | 3890 (97.2%) | 3991 (99.8%) | 4000 |
| | 500–999 | 3988 (99.7%) | 4000 | 4000 | 4000 |
| | 1000+ | 3992 (99.8%) | 4000 | 4000 | 4000 |

Our approach also assumes a uniform sampling of the transmission tree over the time period on which the analysis is carried. It would in theory be prone to under-estimating reporting when entire branches of the transmission tree remain unobserved. For instance, if an epidemic is spreading in a location where surveillance is totally absent, a substantial number of cases may remain unnoticed, and such under-reporting would not be accounted for in our estimates. As a consequence, our method is best applied for estimating the overall reporting over geographic areas and time periods where surveillance has not varied drastically, and which are large enough to yield sufficient case investigations (typically at least 100) for reporting to be accurately estimated. Note that in terms of outbreak response, decisions for altering surveillance strategies would generally be made at coarse geographic scales and considering months of data, so that our method should retain operational value despite its inability to detect changes in reporting at small temporal or spatial scales.

Similarly, the fact that a single overall value of reporting is estimated for the data considered also implies that changes in reporting across different transmission chains will be overlooked. In situations where reporting varies across chains, for instance if super-spreading events are systematically 'better' investigated, the estimated reporting would effectively be an average of the reporting levels of the different chains weighted by their respective numbers of successful case investigations.

We also assumed that the reproduction number ($R$) was independent from the reporting process, so that reported source cases cause the same average number of secondary cases as non-reported ones. This condition may not always be met, for instance if unreported individuals tend to cause more super-spreading events. In the context of Ebola, this may occur through community deaths, in which funeral exposure of a large number of relatives may give rise to a new cluster of cases from a single, unreported source case. Under such circumstances, we would expect our method to under-estimate reporting, although this should be further quantified by dedicated simulation studies.

Another limitation of our method relates to data availability and quality. Our approach relies on case investigation data, a time-consuming but often standard process of contact tracing usually requiring interviews of patients and/or their close relatives. There are several possible outcomes from such investigation: i) identifying a single likely infector amongst reported cases (cases with a known infector) ii) establishing that the infector was not amongst the reported cases (cases without a known infector) iii) failing to identify a single likely infector.

Our approach requires case investigations to fall within the first two categories. In practice, the second and third situations may be easily confused—the third likely being the most frequent. To avoid such confusion, we would recommend recording investigation outcomes as two separate questions: Has a single likely infector been identified? And if yes, is this individual listed amongst reported cases?

In our simulations, we assumed for simplicity that all reported cases were successfully investigated, so that the *reported* outbreak size effectively corresponds to the number of data points available for the estimation. In practice, the actual sample size will be the number of case investigations which led to identifying a single source case (reported, or not). As our method performs better in larger datasets, the data requirement for estimating reporting from transmission chains will involve substantial field work. This also implies that case investigations need to be thorough. Indeed, in situations where the infector has actually been reported, but investigations failed to identify the epidemiological links with their secondary cases, our approach will tend to under-estimate reporting by a factor directly proportional to the frequency of mis-identified links. As a consequence, the proposed methodology is mostly applicable to diseases for which person-to-person transmission can be reliably traced through epidemiological investigation such as EVD. In disease settings where transmission chains are harder to establish, such as COVID-19 where pre-symptomatic and asymptomatic transmission plays an important role, we recommend resorting to other surveillance approaches such as serological surveys to estimate reporting.

Unfortunately, alternative approaches for estimating under-reporting are very demanding in terms of data, typically needing to combine information on dates of onset, location of the cases, full genome sequences of the pathogen for nearly all cases, good prior knowledge on key delays (e.g. incubation period, serial interval) [13,16], and ideally contact tracing data [14]. These methodologies are also much more complex and computer-intensive, as they either involve the reconstruction of transmission trees [13,14] or of outbreak clusters [16]. In contrast, the approach introduced here is fast and simple, and can be used in real time to estimate reporting based on data routinely collected as part of contact tracing activities and surveillance.

We evaluated the performance of the method using simulated EVD outbreaks in line with estimates of transmissibility and epidemiological delays of the Eastern DRC Ebola epidemic [12,28], as this was the original context in which the method was developed. Further work should be devoted to investigating the method's performance for other diseases and different epidemic contexts. In particular, it would be interesting to study the potential impact of correlations between transmissibility and under-reporting, *i.e.* situations in which non-reported cases may exhibit increased infectiousness and cause super-spreading events.

## Conclusion

In this paper, we provided a derivation of a straightforward and pragmatic estimator to real-time estimation of case reporting in outbreak settings, and tested this approach under a range of simulated conditions. The method exhibited little bias, reasonable precision, and while coverage was suboptimal under some settings (in large outbreaks with higher reporting), most estimates were within a reasonable range (10–15%) of the true value. This suggests the method will be useful for informing the response to outbreaks in which person-to-person transmission is the main driver of transmission, and where enough (ideally > 100) chains of transmissions can be retraced through epidemiological investigation.

## Supporting information

**S1 Text.** Table A. Performance measures from 4000 simulation by the mean of the R distribution, reported outbreak size, and true reporting level. Table B. Comparison of absolute error

from 4000 simulations between true reporting levels and estimate of reporting by the mean of the R distribution, reported outbreak size, and true reporting level.
(DOCX)

## Author Contributions

**Conceptualization:** Christopher I. Jarvis, Flavio Finger, Tim P. Morris, Jennifer A. Thompson, Olivier le Polain de Waroux, W. John Edmunds, Sebastian Funk, Thibaut Jombart.

**Formal analysis:** Christopher I. Jarvis, Amy Gimma, Flavio Finger, Tim P. Morris, Jennifer A. Thompson, W. John Edmunds, Sebastian Funk, Thibaut Jombart.

**Funding acquisition:** W. John Edmunds.

**Investigation:** Christopher I. Jarvis, Amy Gimma, Flavio Finger, Tim P. Morris, Jennifer A. Thompson, Olivier le Polain de Waroux, W. John Edmunds, Thibaut Jombart.

**Methodology:** Christopher I. Jarvis, Amy Gimma, Flavio Finger, Tim P. Morris, Jennifer A. Thompson, Olivier le Polain de Waroux, W. John Edmunds, Sebastian Funk, Thibaut Jombart.

**Software:** Christopher I. Jarvis, Amy Gimma, Thibaut Jombart.

**Supervision:** W. John Edmunds, Sebastian Funk, Thibaut Jombart.

**Validation:** Christopher I. Jarvis, Amy Gimma, Tim P. Morris, Jennifer A. Thompson, Olivier le Polain de Waroux, Sebastian Funk, Thibaut Jombart.

**Visualization:** Christopher I. Jarvis, Amy Gimma, Thibaut Jombart.

**Writing – original draft:** Christopher I. Jarvis, Amy Gimma, Thibaut Jombart.

**Writing – review & editing:** Christopher I. Jarvis, Flavio Finger, Tim P. Morris, Jennifer A. Thompson, Olivier le Polain de Waroux, W. John Edmunds, Sebastian Funk, Thibaut Jombart.

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
