## [Decision Letter · Decision Letter 0]

5 Mar 2021

Dear Dr Jombart,

Thank you very much for submitting your manuscript "Measuring the unknown: an estimator and simulation study for assessing case reporting during epidemics" for consideration at PLOS Computational Biology.

As with all papers reviewed by the journal, your manuscript was reviewed by members of the editorial board and by several independent reviewers. In light of the reviews (below this email), we would like to invite the resubmission of a significantly-revised version that takes into account the reviewers' comments.

We cannot make any decision about publication until we have seen the revised manuscript and your response to the reviewers' comments. Your revised manuscript is also likely to be sent to reviewers for further evaluation.

Sincerely,

Benjamin Muir Althouse

Associate Editor

PLOS Computational Biology

Tom Britton

Deputy Editor

PLOS Computational Biology

Reviewer's Responses to Questions

**Comments to the Authors:**

Reviewer #1: The authors here seek to solve a critical problem in outbreak settings, the difficulty in understanding the reporting fraction in near-real time, without costly and time-consuming serological and genomic studies. The approach is simple (assuming extensive contact tracing and case investigation data exist) and intuitive, and the authors have structured the paper clearly.

My comments below seek to improve an approach for which I have high enthusiasm and believe could be very useful to the field. More clarity in the derivation/formalism and more extensive sensitivity analyses are need to show the validity of the approach, and several parts of the approach and simulation should be tailored to better address real-world outbreak scenarios (e.g., time-varying reporting, shorter time scale simulations).

1. I was confused whether this method accounts for undetected secondary cases. It seems the expression E(n_u + n_k) = (m_r+m_K)*R would imply complete detection of the secondary cases. The expressions for E(n_k) and E(n_u), though, both have a pi term that I believe is account for under-reporting of secondary cases. Why is that not present in E(n_u + n_k)?

Perhaps time indexing the estimator pi would help resolve some confusion. My understanding is that:

pi_t = m_{r,t} / (m_{r,t} + m_{u,t})

n_{t+1} = (m_{r,t} + m_{u,t})*R (where n_{t+1} is the total number of secondary cases caused by index cases in generation t)

n_{u, t+1} = m_{u,t}*R

n_{r, t+1} = m_{r,t}*R

n_{u, detected, t+1} = m_{u,t}*R*pi_{t+1}

n_{r, detected, t+1} = m_{r,t}*R*pi_{t+1},

where I've replaced n_u from the paper with n_{u, detected, t+1}, the number of detected secondary cases infected by an unknown infector of the previous generation, t, and n_k with n_{r, detected, t+1} for the number of detected secondary cases infected by a known infector of generation t. The derivation then relies on the assumption that, at least over the interval of a generation time, pi_t ~ pi_{t+1}. Does this fit with the authors' logic? I think the assumption pi_t ~ pi_{t+1} would be mostly valid in many scenarios, but should be explicitly stated.

2. Time-indexing the reporting fraction would help indicate whether or how this method can be used to estimate a time-varying reporting fraction, as reporting fractions seem rarely to be constant (and cause the must frustration when they are not constant). This would be a critical expansion of the methods, and additional simulations with a non-constant reporting fraction would improve the paper.

This formalism could also help clarify to which time interval estimates apply. That is, should we consider pi_hat = n_k / (n_k + n_u), to use the original notation, as a lagged estimator by one generation time? How do reporting delays factor in? It's a fine point, but given recent discussion of cohort vs case reproductive numbers in this journal (https://doi.org/10.1371/journal.pcbi.1008409) and the relationship between time-varying reporting and R_t, perhaps of interest to the intended audience.

3. I am concerned about the situation where an infector is reported, but not known to be the infector of a given case. That is, what happens when the assumption that "all cases reported were investigated, so that it is known if they had a documented epidemiological link, or not, amongst reported cases" breaks down? I understand this may be less common in the context of EVD, but seems more likely in the case of COVID, for example, for which this method would be very useful. In this situation, the unknown but detected infector would lead to inflation of the denominator (i.e., over-counting of n_u) and underestimation of pi. Some discussion or simulation (e.g. by pruning known transmission links, not solely index cases) would be useful.

This would be a larger expansion, but it does seem like it would be possible to incorporate some probabilistic reconstruction (rather than just known/unknown links) to account for scenarios of several likely infectors, or an unknown infector, as described in the discussion.

4. I can't speak as much to the theoretical validity of using standard errors for a proportion and exact binomial confidence intervals, though I am concerned by the low coverage. My two primary concerns, both partially addressed in the limitations, are (1) possible links/correlation of detection in a transmission chain (i.e., a secondary case of an undetected index is less likely to be detected itself; it would seem this might have less influence on uncertainty and may be more of a structural issue, that pi_{t+1} depends on m_{u,t} vs m_{r,t}) and (2) when there is relationship between R and pi (e.g., super-spreading events more likely to be detected), or even just overdispersion in R leading to larger variance in n and hence pi.

The discussion does partially address #1, saying the approach would "be prone to under-estimating reporting when entire branches of the transmission tree remain unobserved"; but what if there is not complete non-report of certain chains, and just systematic under-reporting?

Similarly, the authors note that if super-spreaders were more likely to be undetected, "we would expect our method to under-estimate reporting, although this should be further quantified by dedicated simulation studies". Are the authors not able to perform these simulations? Does simulacr allow for negative binomial branching?

5. The definitions of bias and coverage are very clear. I was confused, though, by the statement that "the model based standard error is mean of the square of the bias", when Table 2 lists the model based standard error as the square root of Var(pi). How are the authors defining Var(pi) (presumably distinct from the standard error of pi_hat on pg 6?)

6. It seems all models were run for 365 days with a fixed distribution of R, and thus it was differences in population size and reporting that dictated reported outbreak size. It would greatly improve the paper to see the patterns in bias/precision broken down by R and time scale, not just a proxy of population size. That is, does a reported outbreak of n=100 across one year due to low R and small population behave the same as a reported outbreak of n=100 across one month with higher R? Exploring these shorter time frames would seem to be highly relevant for the outbreak scenarios in which this method would be applied. I also suspect in short time scales and smaller populations are where differences in a Poisson vs negative binomial branching process (or even just differences in the distribution of R) could become most important in appropriately considering uncertainty.

Reviewer #2: The authors describe a novel method for estimating the proportion of cases that are observed during an epidemic. Overall the manuscript is well presented and the methods and results described clearly.

I raise (major) points below about the appropriateness of the measure used to estimate reporting (cases with "known" infectors) and the generalisability of their findings across outbreaks of different pathogens.

1. The key measure from the epidemiological data used is "cases with a known infector". I initially looked for information on how a "known infector" is classified in the Methods, but found it later in the Discussion. "Our approach relies on case investigation data, a time-consuming process usually requiring interviews of patients and/or their close relatives. There are several possible outcomes from such investigation: i) a single infector can be identified..." etc

As this definition and the description of the underlying data is important it should be visible much earlier in the paper.

My impression is that determining "who infected whom" is difficult in the absence of genomic data. Contact tracing methods may be effective when a pathogen is "rare", i.e. a primary infection occurs in a village in a person who has travelled from outside, and then a second case is reported in the same household within the serial interval. But such examples are likely rare?

When transmission is widespread and there are multiple transmission chains circulating simultaneously, inferring who infects whom is very difficult. Therefore the accuracy of interviews to determine infector/ infectee pairs is likely to decrease as the outbreak size increases. Do the authors agree that this is a potential source of bias that should be accounted for in their simulations?

2. It is implied that the method is generalisable across outbreaks with different pathogens. In using data only from a single pathogen in a single location, however, this claim is suspect and not proven.

Could simulations be performed which are calibrated to the natural history of other pathogens? In particular, I wonder if the method would be successful at inferring the level of reporting in infections with a high proportion of asymptomatic cases or with a variable serial interval. In both of these cases, I suspect it would be more difficult to correctly assign infector/ infectee pairs.

3. The underlying model does not account for an over-dispersed distribution of secondary cases (negative binomial distribution, or "super-spreading"). Although this is mentioned as a limitation in the Discussion, it does raise questions about the appropriateness of the epidemiological model, given that overdispersion in the secondary case distribution is a key driver of epidemics. There are estimates of the dispersion parameter k for Ebola virus (e.g. 0.37 in Lau et al. PNAS 2017), therefore the authors could use this information in their model. For instance, if one person is identified as an infector, what is the probability that they did indeed only infect one other person? When k is small, this probability is low and there are likely several other cases non-reported.

4. In the introduction the authors characterise alternative measures to estimate the reporting rate as difficult or impractical, although the data collecting method for their own analysis is later described as "a time-consuming process". There is no mention of prevalence surveys to estimate the true number of cases - e.g. the REACT study which is ongoing in the UK for COVID-19. How would (a small number of) cluster-randomised prevalence surveys perform as a method to estimate reporting, given that interviewing cases is "time consuming", and likely subject to bias (see point 1)?

**Have all data underlying the figures and results presented in the manuscript been provided?**

Reviewer #1: Yes

Reviewer #2: Yes

PLOS authors have the option to publish the peer review history of their article (what does this mean?). If published, this will include your full peer review and any attached files.

Reviewer #1: No

Reviewer #2: No
---

## [Decision Letter · Decision Letter 1]

19 Sep 2021

Dear Dr. Jarvis,

Thank you very much for submitting your manuscript "Measuring the unknown: an estimator and simulation study for assessing case reporting during epidemics" for consideration at PLOS Computational Biology.

As with all papers reviewed by the journal, your manuscript was reviewed by members of the editorial board and by several independent reviewers. In light of the reviews (below this email), we would like to invite the resubmission of a significantly-revised version that takes into account the reviewers' comments.

We cannot make any decision about publication until we have seen the revised manuscript and your response to the reviewers' comments. Your revised manuscript is also likely to be sent to reviewers for further evaluation.

Sincerely,

Benjamin Muir Althouse

Associate Editor

PLOS Computational Biology

Tom Britton

Deputy Editor

PLOS Computational Biology

Reviewer's Responses to Questions

**Comments to the Authors:**

Reviewer #1: I thank the reviewers for their thorough responses to my comments. This paper would continue to benefit from refinement of how the methods and its valid applications are presented, as discussed below.

1. I still find there to be a bit of an internal conflict in the described applications of this method. The method is described in the abstract and introduction as "useful for estimating reporting in real-time" and as a tool that can be used "to inform strategic decision making during an outbreak response". The proposed use case in the author's response, however, is to guide decisions for altering surveillance strategies using "months of data", rendering concerns about temporal and spatial variations of lesser importance. To me, that's not exactly a 'real-time' application and doesn't allow for operational results in the critical early weeks or months (depending on case rate) of an outbreak. Relatedly, while I think a popular use of reporting fraction estimates is to understand current burden (think of the extensive "nowcasting" COVID literature), this method would appear to be ill-suited except in the case that extensive investigation of multiple parallel transmission chains can be completed quickly.

Clarifying the valid uses of this method earlier in the paper, specifically the sample size limitations and associated limitations in temporal and spatial resolution and the bias from incomplete tracing, would improve the framing of this paper. I would soften language in the introduction, particularly in paragraph 2 when describing "timely estimation" amidst changing outbreak situations.

2. Thank you to the authors for providing the reference for their definitions of model based standard error and empirical standard error. The definition of the empirical SE is now more clear. I am still confused, though, by the definition of the model-based standard error. The definition E[(theta_hat - theta)^2] given in Table 3 is the classical definition of the mean squared error; this is also the name used in Table 6 of Morris et al. The written definition that the model-based square error (the MSE) is "the mean of the square of the bias" is confusing, in that in that MSE = E[Bias_i^2], if Bias_i = theta_{hat,i} - theta, but MSE!=Bias^2, when Bias = E(theta_hat)-theta, as defined earlier in Table 3. Did the authors intend to use a different quantity than the MSE? If not, it would be more clear and better match common terminology to refer to the model-based square error as the MSE.

If the model standard error is in fact the MSE, I would expect MSE = Var(theta_hat) + Bias^2, where Bias = E(theta_hat)-theta. This relationship does not appear to hold in Table 3 (e.g., for pi=0.25 and outbreak size 10-99, I would expect MSE ~ 0.071^2 = 0.005). The derived quantity in the authors' code may be incorrect. The model based standard error (`mod_se_bias`, or `model_se` as used in Table 3) appears to be the square root of the mean across all simulations of the squared reporting probability standard errors [that is, sqrt( mean( reporting_probability * (1 - reporting_probability) / n_reported ) )]. What is defined as `rmse` or `root_mean_squared_error` appears to be correct [that is, sqrt( mean( (reporting_fraction - true_value)^2 ) )], but does not appear to be used in Table 3.

3. Do the authors mean "ii) established that the infector was NOT amongst the reported cases (cases without a known infector)."? I think it is worth addressing the difference between establishing that the infector was not amongst reported cases and being unable to establish whether the infector is among reported cases; the former implies more certainty than I think there often is in the case of unidentified infectors.

4. The sentence "As a consequence, the proposed methodology is mostly applicable to diseases for which person-to-person transmission can be achieved through epidemiological investigation such as EVD" is unclear; do the authors mean that "person-to-person transmission can be reliably traced" or similar? This is a critical assumption that merits more treatment in the methods (e.g., following the definitions of cases with/without known infectors in "Estimating reporting from epidemiological links").

5. I would explicitly state in the derivation that reported and unreported infectors are assumed to have same distribution of R (or do the authors believe that only the average R must be constant, in which case this should be stated).

6. I believe the definition of the exact (Clopper Pearson) interval includes the multiplier (n_k + 1) in the denominator of the second term, rather than n_k alone, for the upper bound. The correct definition was implemented in the code.

7. Code for reproducing simulations and manuscript figures is generally well commented. This could be user error, but I was unable to run the parallelized outbreak simulations without adding the pkg:: operator whenever `simulacr` functions were required (simulacr::make_disc_gamma in create_raw_simulation_list and simulacr::simulate_outbreak in recursive_minimum_outbreak, specifically). Better documentation for which version/branch to use and/or a code release would be helpful.

Reviewer #2: The authors have provided robust responses to the reviewer comments and made modest alterations to their manuscript.

My remaining concern is that the method has not been applied to estimate reporting in the original EVD dataset. To pose as a question - what was the fraction of cases reported during the EVD epidemic in DRC? Without this application the manuscript feels incomplete, particularly as there are references in both the paper and the response to reviewers that this study is EVD focussed rather than a generalisable tool. It also undermines the point made in the Introduction that the fraction of cases reported is a "key epidemiological indicator" and an "essential factor to consider". If reporting the level of reporting is so important, then why not report it? As the method is "fast and simple" this should presumably not take long.

In addition, some context on the original dataset would be beneficial. There are pointers to other studies, but the Methods should include, as a minimum, the locations where contact tracing data were collected, the dates of collection, and by whom they were collected (Public health ministry/ NGO). Ideally a template of the interview form used for contact tracing should be included as supplementary information.

**Have the authors made all data and (if applicable) computational code underlying the findings in their manuscript fully available?**

Reviewer #1: Yes

Reviewer #2: None

PLOS authors have the option to publish the peer review history of their article (what does this mean?). If published, this will include your full peer review and any attached files.

Reviewer #1: No

Reviewer #2: No
---

## [Decision Letter · Decision Letter 2]

20 Apr 2022

Dear Dr. Jarvis,

We are pleased to inform you that your manuscript 'Measuring the unknown: an estimator and simulation study for assessing case reporting during epidemics' has been provisionally accepted for publication in PLOS Computational Biology.

Best regards,

Benjamin Althouse

Associate Editor

PLOS Computational Biology

Tom Britton

Deputy Editor

PLOS Computational Biology

Reviewer's Responses to Questions

**Comments to the Authors:**

Reviewer #1: I thank the reviewers again for their thoughtful responses and edits. I have no further comments. This is an excellent paper - congratulations to all.

Reviewer #2: The study is insufficient in its current form as the main parameter of interest, the proportion of cases reported, is not inferred from contact tracing data. It is not sufficient to use the method only on simulated data given the epidemiological focus of the manuscript, and the method is not interesting enough by itself to warrant publication in PLoS Comp Biol. The authors need to apply their method to at least one real world dataset to demonstrate that i) it is feasible to perform with routinely collected data from case investigations, and ii) that estimating "reporting" is insightful for outbreak response.

**Have the authors made all data and (if applicable) computational code underlying the findings in their manuscript fully available?**

Reviewer #1: Yes

Reviewer #2: None

PLOS authors have the option to publish the peer review history of their article (what does this mean?). If published, this will include your full peer review and any attached files.

Reviewer #1: No

Reviewer #2: No

---

## [Editor Report · Acceptance letter]

17 May 2022

PCOMPBIOL-D-21-00139R2 

Measuring the unknown: an estimator and simulation study for assessing case reporting during epidemics

Dear Dr Jarvis,

I am pleased to inform you that your manuscript has been formally accepted for publication in PLOS Computational Biology. Your manuscript is now with our production department and you will be notified of the publication date in due course.

With kind regards,

Anita Estes
